# The Immune Microenvironment in Pancreatic Cancer

**DOI:** 10.3390/ijms21197307

**Published:** 2020-10-03

**Authors:** Magdalena Huber, Corinna U. Brehm, Thomas M. Gress, Malte Buchholz, Bilal Alashkar Alhamwe, Elke Pogge von Strandmann, Emily P. Slater, Jörg W. Bartsch, Christian Bauer, Matthias Lauth

**Affiliations:** 1Institute for Medical Microbiology and Hospital Hygiene, Philipps University Marburg, 35043 Marburg, Germany; magdalena.huber@staff.uni-marburg.de; 2Institute of Pathology, University Hospital Giessen-Marburg, 35043 Marburg, Germany; brehmc@med.uni-marburg.de; 3Department of Gastroenterology, Endocrinology, Metabolism and Infectiology, Center for Tumor- and Immunology (ZTI), Philipps University Marburg, 35043 Marburg, Germany; gress@med.uni-marburg.de (T.M.G.); malte.buchholz@staff.uni-marburg.de (M.B.); christian.bauer@med.uni-marburg.de (C.B.); 4Institute for Tumor Immunology, Clinic for Hematology, Oncology and Immunology, Center for Tumor Biology and Immunology (ZTI), Philipps University Marburg, 35043 Marburg, Germany; poggevon@staff.uni-marburg.de (E.P.v.S.); bilal.alashkaralhamwe@uni-marburg.de (B.A.A.); 5Department of Visceral-, Thoracic- and Vascular Surgery, Philipps University Marburg, Baldingerstrasse, 35043 Marburg, Germany; slater@med.uni-marburg.de; 6Department of Neurosurgery, Philipps University Marburg, Baldingerstrasse, 35043 Marburg, Germany; jbartsch@med.uni-marburg.de

**Keywords:** pancreatic cancer, tumor stroma, T-cells, natural killer cells, macrophages, neutrophils, cancer-associated fibroblasts, immunotherapy

## Abstract

The biology of solid tumors is strongly determined by the interactions of cancer cells with their surrounding microenvironment. In this regard, pancreatic cancer (pancreatic ductal adenocarcinoma, PDAC) represents a paradigmatic example for the multitude of possible tumor–stroma interactions. PDAC has proven particularly refractory to novel immunotherapies, which is a fact that is mediated by a unique assemblage of various immune cells creating a strongly immunosuppressive environment in which this cancer type thrives. In this review, we outline currently available knowledge on the cross-talk between tumor cells and the cellular immune microenvironment, highlighting the physiological and pathological cellular interactions, as well as the resulting therapeutic approaches derived thereof. Hopefully a better understanding of the complex tumor–stroma interactions will one day lead to a significant advancement in patient care.

## 1. Pancreatic Cancer: Clinical Situation

Pancreatic ductal adenocarcinoma (PDAC) exhibits the poorest prognosis of all solid tumors, with a median survival of 6 months and steadily increasing incidence rates in the industrialized world [1]. This malignancy is projected to become the second leading cause of cancer death by 2030 in the US [2]. Clinical diagnosis is often delayed due to the relatively unspecific clinical symptoms, such as back pain, a loss of appetite, a loss of weight, and a new onset of diabetes, and thus early diagnosis is rare [3]. Specialized imaging modalities, such as high-resolution ultrasound, endoscopic ultrasound (EUS), computed tomography (CT), and magnetic resonance imaging (MRI), are required to reach a definite diagnosis. Upon diagnosis, >80% of patients have either locally advanced disease or distant metastases, mostly to the liver, that render surgical intervention impossible [4]. A total of 15–20% of patients can be resected with complete removal of the pancreatic tumor; however, local or distant recurrence is often observed within the first 2 years after surgery. Recently, the survival of patients with resectable tumors was shown to be significantly improved by neoadjuvant radiochemotherapy [5] and to an impressive 54-month median overall survival with adjuvant FOLFIRINOX chemotherapy [6]. FOLFIRINOX has subsequently become the standard of care in the adjuvant treatment of patients with resectable pancreatic cancer.

Systemic chemotherapy is the only treatment option for patients with advanced, metastatic PDAC. Most patients who present with advanced disease die within 12 months of diagnosis as many of them have chemo-refractory disease [7]. Only in the last decade has a significant survival extension been achieved with combination chemotherapies such as FOLFIRINOX [8] and Gemcitabine plus Nab-Paclitaxel [9] for first-line therapy, and the combination of 5-FU and folinic acid plus a liposomal formulation of irinotecan (Nal-Iri) as second-line treatment [10]. However, since only a fraction of patients will respond to these individual regimens, the development of individualized, targeted therapies is of paramount importance. However, the Genome Project has revealed that PDAC is a tumor with a high degree of inter-tumoral genetic heterogeneity [11], suggesting that no single targeted therapy will work for all PDAC patients. PDAC molecular subtypes have recently been described, and a consensus is being reached that two major transcriptomic subtypes of PDAC exist (“classical” and “basal-like”), which are of biological, prognostic, and possibly predictive relevance [12,13,14,15,16]. Patients with basal-like PDACs seem to respond poorly to established chemotherapy protocols [12,13]. By digitally separating tumor, stromal, and normal gene expression, Moffitt and colleagues, in addition to confirming the two ‘basal-like’ and ‘classical’ tumor subtypes, defined ‘normal’ and ‘activated’ stromal subtypes, which they suggested to be independently prognostic [16]. Patients whose samples belonged to the activated stromal subtype had a worse median survival time when compared to patients whose samples belonged to the normal stromal subtype (median survival time of 15 vs. 24 months). They found that activated stroma was characterized by a more diverse set of genes associated with macrophages, such as the integrin *ITGAM* and the chemokine ligands *CCL13* and *CCL18*. Activated stroma also expressed *FAP* (encoding the fibroblast activation protein), which has previously been related to a worse prognosis, and suggests that an activated fibroblast state may be partially responsible for the poor outcomes for patients with this stromal subtype [16].

PDAC driver mutations have proven difficult to target in the clinical setting, with the exception of microsatellite instability with immune check point inhibitors [17] or *BRCA1/2* mutations with platinum-based chemotherapies and PARP-inhibitors [18,19]; however, these concern only a small number of patients. *KRAS* wild-type patients comprise between 5 and 8% of sporadic PDAC patients [20] and have been shown to harbor actionable genomic alterations [21], such as *NTRK* [22] or *NRG1* [23,24] fusions. While molecular subtypes and actionable genomic alterations may theoretically help guide precision medicine approaches, the molecular characterization of PDAC in patients with advanced disease has not yet entered routine clinical practice. Biopsy-driven genomic and drug screening studies have been challenging due to rapid disease progression and the small-volume and heterogeneous nature of biopsies that impede deep molecular characterization. Only recently were the first promising results of biopsy-based transcriptomic subtype analyses with limited patient numbers published [12,13,20,25,26]. ASCO [27] and NCCN (NCCN Guidelines Version 1.2020) have recently suggested performing germline and somatic gene testing with panel sequencing in all patients presenting with advanced sporadic PDAC. These recommendations will currently only benefit 5–10% of sporadic PDAC patients and are not yet based on prospective evidence and a similar consensus has not yet been achieved elsewhere (e.g., [28]).

To summarize, in contrast to other tumor entities such as melanoma, lung cancer, and breast cancer, medical therapeutic innovations have been scarce in PDAC and survival rates have only marginally improved over the last few decades [1].

Histologically, PDAC is characterized by a dense stromal architecture with massive extracellular matrix (ECM) deposition, rendering this tumor entity one of the most stroma-rich solid tumor types. The tumor cell-centric view of previous decades has probably contributed to the lack of significant progress in successful drug development for pancreatic cancer. It is now undisputed that the stroma is a defining feature of this disease, regulating central processes such as tumor growth, vascularization, drug responsiveness, and metastasis. As such, the tumor microenvironment itself has become a target of today’s drug development efforts. In this review, we will describe currently available knowledge on major cellular components of the PDAC stroma, starting with innate immune cells such as macrophages, NK cells, and neutrophils, and followed by adaptive T cell-mediated immunity. Furthermore, cancer-associated fibroblasts (CAFs) will be discussed as important mediators of the PDAC immune landscape. When concluding this review, we will outline current activities in the emerging field of immunotherapy, which is trying to translate tumor microenvironment (TME) knowledge into a clinical benefit for patients.

## 2. Role of Innate Immune Cells in PDAC: Macrophages and Myeloid-Derived Suppressor Cells (MDSCs)

Innate immune cells of the myeloid lineage, including granulocytes, monocytes, macrophages, and dendritic cells, play an important role in cancer cell recognition, the initiation of inflammation, and antitumor responses [29]. Tumor cells, however, often develop mechanisms to evade immune surveillance, and persistent inflammation has been shown to be a driver of tumor progression in many malignancies, including pancreatic cancer [30]. Myeloid cells thus play a dual role in cancer, on the one hand initiating antitumor responses, but also promoting local inflammation leading to chronic cancer-associated inflammation [31,32].

### 2.1. The Role of Macrophages in PDAC

The majority of macrophages in healthy and inflamed tissues differentiate from bone marrow-derived monocytes in the peripheral blood circulation, although specialized tissue-resident macrophages that are not derived from blood monocytes, such as alveolar macrophages in the lungs, microglia in the brain, and Kupffer cells in the liver, exist [33]. Through the presence of chemokines, cytokines, and other secreted factors (e.g., GM-CSF, CSF-1, IL-3, CXCL12, and CCL2), as well as other environmental clues, such as local anoxia and high levels of lactic acid, circulating monocytic cells are recruited to the tumor microenvironment and become tumor-associated macrophages (TAMs) [34,35]. It should be noted, however, that recent evidence suggests that TAMs may also derive from tissue-resident macrophages, possibly representing a functionally distinct subpopulation of TAMs [36]. TAMs display different functional states, termed polarization states, during tumor initiation, progression, and therapeutic intervention. A large and ever-increasing spectrum of TAM subpopulations has been described, which is commonly broadly divided into “M1” and “M2” macrophages to characterize the polar opposite extremes of a continuous spectrum of polarization states. In general terms, M1 macrophages have been described to secrete pro-inflammatory cytokines with predominantly anti-neoplastic effects, while M2 macrophages produce anti-inflammatory signals which may facilitate tumor progression [37,38,39] (Figure 1).

A number of studies have demonstrated an inverse correlation between patient prognosis and TAM infiltration in various tumor entities, including pancreatic cancer [40,41,42,43]. Functionally, this is explained by a number of different effects. Using mouse models of PDAC, different groups have demonstrated that TAMs mediate immunosuppression and angiogenesis and promote tumor progression by releasing cytokines, proteases, and growth factors such as VEGF [44,45,46,47]. Moreover, the depletion of macrophages attenuated metastasis to the lung and liver in genetic [45] and orthotopic [48] mouse models of pancreatic cancer, and inflammatory monocytes have been described to facilitate liver metastasis formation by promoting stellate cell-mediated fibrosis in the liver [49].

In addition, TAMs are able to profoundly impair the efficacy of chemotherapy in PDAC. TAMs have been shown to influence the activity of cytidine deaminase (CDA), which is a key metabolizer of gemcitabine, thereby driving resistance to gemcitabine-based chemotherapy in in vivo PDAC models [50]. Moreover, TAMs impair the efficacy of therapeutic irradiation and targeted antiangiogenic treatment by promoting angiogenic escape mechanisms (for a recent review, see [51]). Consequently, CCR2 inhibition has been demonstrated to improve the efficacy of chemotherapy, inhibit metastasis, enhance the efficacy of radiotherapy, and increase T cell immune infiltration by blocking monocyte recruitment to the tumor microenvironment in mouse models of PDAC [46,48,52,53]. TAMs can thus be effectively targeted for therapeutic benefit in preclinical PDAC models, making them potential targets for novel therapeutic strategies in human PDAC.

### 2.2. The Role of Myeloid-Derived Suppressor Cells

Myeloid-derived suppressor cells (MDSCs) are a heterogeneous population of immature myeloid cells that are commonly divided into two groups of cells termed granulocytic or polymorphonuclear (PMN-MDSC), which are phenotypically and morphologically similar to neutrophils, and monocytic (M-MDSC), which are phenotypically and morphologically similar to monocytes [54,55,56]. MDSCs are strongly increased in number in both the circulation and the microenvironment of human tumors, and typically, PMN-MDSCs represent more than 80% of all tumor-associated MDSCs [55].

In human PDAC, MDSC levels correlate with the clinical cancer stage [57,58,59]. Results from genetically engineered mouse models suggest that the granulocyte-macrophage colony-stimulating factor (GM-CSF), produced by tumor cells from early cancer stages onwards, is a major factor stimulating MDSC recruitment and differentiation [60,61]. In addition, the receptor for advanced glycation end products (RAGE) is overexpressed in both human PDACs and GEMMs and its ablation was associated with a decreased MDSC frequency and delayed pancreatic carcinogenesis [62]. Similarly, the yes-associated protein (Yap) has been described to contribute to an immunosuppressive tumor microenvironment by inducing the expression and secretion of cytokines and chemokines involved in the differentiation and accumulation of MDSCs [63]. Within tumor tissue, the expansion of MDSC numbers has been reported to be driven by elevated CD200 expression [64].

In the tumor, MDSCs contribute to suppressing CD4^+^ and CD8^+^ T cell function. One mechanism occurs through direct cell-cell contact of MDSCs with lymphocytes, which leads to the upregulation of programmed death-ligand 1 (PD-L1) by MDSCs. The PD-L1/PD-1 interaction, in turn, leads to the suppression of T-cell activation and self-tolerance [65]. In addition, MDSCs have been shown to stimulate the expansion of immunosuppressive regulatory T cells (Tregs) through the IL-10-dependent secretion of TGF-β and IFN-γ [66,67], again resulting in suppression of the T cell function. Consequently, the targeted depletion of MDSCs in GEMMs by different experimental strategies has been demonstrated to activate an effective endogenous antitumor T cell response in established tumors and to significantly impair Kras^G12D^-driven pancreatic cancer initiation when induced at early stages [68,69].

In addition to their role in blunting the anti-tumor response of the adaptive immune system, MDSCs also have the ability to attenuate innate anti-tumor immunity through a variety of mechanisms. Among others, they have been shown to inhibit natural killer cell (NK cell) cytotoxicity through cell-contact-dependent mechanisms and promote the conversion of macrophages towards an M2 phenotype in model systems of lung and breast cancer [70,71,72]. Taken together, an increasing body of evidence suggests that MDSCs are central inhibitors of anti-tumor immunity, emphasizing their attractiveness as targets for novel therapeutic concepts.

## 3. Contribution of Natural Killer (NK) Cells in PDAC

NK cells represent 5–20% of circulating lymphocytes in humans [73,74]. This subtype of innate lymphoid immune cells is derived from CD34^+^ hematopoietic progenitor cells in the bone marrow and defined by the lack of surface T cell receptors (TCRs), the expression of the neural cell adhesion molecule (NCAM; also known as CD56), and the natural cytotoxicity receptor (NCR) NKp46 [75]. NK cells were originally identified as unspecific killer cells that recognize and directly kill virus-infected or tumor cells without prior antigen stimulation via the main cytotoxicity receptors CD16, NKG2D, DNAM-1, and NCRs. Therefore, NK cells are a critical immune component in controlling tumor growth, as demonstrated in different experimental mouse models [76,77].

Later, it was recognized that NK cells are also regulatory cells able to shape the anti-tumor immune response by reciprocal interactions with dendritic cells, macrophages, T cells, and endothelial cells through a combination of cell surface receptors and secreted cytokines [73,78]. However, recent data suggest that mediators within the tumor microenvironment (TME), including soluble factors or tumor cell-derived extracellular vesicles (EVs), can induce functional alterations in NK cells, which circumvent recognition and elimination by NK cells and additionally polarize them towards a tumor-promoting phenotype [79,80] (Figure 2).

Many hypotheses leading to an immune escape of tumor cells from NK cells have been proposed. Among others are the expression of MHC-I molecules, the downregulation of activating NK receptors, and the shedding of NKG2D ligands, all of which impair the NK cell function [81,82,83,84]. The activity and role of NK cells in anti-tumor immune responses in PDAC are only partially understood.

The absolute number of NK cells in the circulation was positively correlated with survival in PDAC patients [85]. However, the cytotoxicity of PDAC-associated NK cells was impaired compared to those of healthy individuals [85]. This was reflected by their reduced production of the cytotoxic granule components granzyme B and perforin, which are key mediators in the elimination of cancer cells [86]. Furthermore, PDAC has recently been shown to mediate immune suppression by the expression of several mediators, including transforming growth factor beta (TGF-β), interleukin (IL)-10, indoleamine 2,3-dioxygenase (IDO), and matrix metalloproteinases (MMPs), which impair NK cell tumor cell recognition and killing, e.g., via the downregulation of cytotoxicity receptors (Peng 2014). A reduced expression of these receptors, including NKG2D, NKp46, and NKp30, was observed for peripheral NK cells from PDAC patients and this correlated with the stage and histological grade of the patients [87,88]. Another study reported that cancer progression in PDAC patients correlated with the downregulation of the nectin-like binding molecules DNAX accessory molecule (DNAM)-1 and Tactile receptor (CD96) on NK cells, which are additional receptors involved in target cell recognition [89]. An additional example reflecting NK cell inhibition in PDAC is the secretion of the Igγ-1 chain C region (IGHG1), which has the ability to bind competitively on the Fcγ receptor of NK cells, subsequently reducing the antibody-dependent cell-mediated cytotoxicity (ADCC) [90]. Aside from that, it has been found that the abundance of activated pancreatic stellate cells in the stroma of pancreatic cancer can diminish NK cell functions in the local tumor microenvironment [91]. Moreover, it was demonstrated that PDAC-NK cells exhibit an impaired tumor localization that was associated with the downregulation of the chemokine receptor CXCR2 [92]. Therefore, PDAC tumors show an extremely low frequency of NK cell infiltration (<0.5%) counteracting tumor cell elimination. Of note, the in vitro proliferation rate of isolated tumor-infiltrating NK cells was associated with a longer overall survival [92].

Several cytokines have been studied that either contribute to NK cell activation, such as IL-12, IL-15, and IL-2, or that inhibit NK cell functions, such as IL-18, IL-10, and TGF-β. TGF-β, produced by tumor cells and immune cells, impairs NK cell function directly or indirectly by cell-cell contact [93,94]. Among the direct effects are the downregulation of activating receptors, including NKp30 and NKG2D [95,96], which is observed for peripheral PDAC-NK cells. Interestingly, TGF-β is able to transdifferentiate NK cells into group 1 innate lymphoid cells (ILC1), which lack cytotoxic functions [97]; however, their role in PDAC has not been addressed so far. Of note, PDAC is associated with higher levels of TGF-β and IL-10, locally in the tumor tissue and systemically in the blood, supporting immune evasion [98]. Moreover, the dysfunction of NK cells was associated with the level of TGF-β, which, in turn, correlated with cancer recurrence and high-grade tumors [99].

However, as described for other cancers [100], it seems possible to restore the restricted cytotoxicity of tumor-associated NK cells upon ex vivo stimulation and proliferation. Of note, the tumor cell killing efficacy mediated by these patient-derived NK cells was comparable to the efficacy of NK cells isolated from healthy donors [92], drawing interest to immune therapies aimed at restoring NK cell functions [101,102]. In the same line, it was reported that ex vivo expanded and antibody-activated NK cells that were adoptively transferred to patients exhibited potent IFNγ secretion and significant anti-tumor activity against pancreatic cancer cells [103].

Another study using a mouse model of pancreatic cancer has shown that the incubation of NK cells with Hsp70-peptide (TKD), in combination with low-dose IL-2, enhances the cytolytic and proliferative capacity of NK cells against Hsp70 surface-positive tumors [104]. The combination of the NK cell-activating interleukin IL-15 and therapeutic antibodies, which engaged the co-stimulatory receptor CD40, improved anti-tumor responses in Panc02 or KPC mouse tumor models [105]. The therapeutic effect was mediated by T cells and NK cells. Moreover, administration of the NK cell-recruiting protein (NRP)-conjugated antibody (NRP-body) increased NK cell infiltration into tumor tissue and reduced the tumor burden [106]. Additionally, two clinical trials have illustrated that the combination of allogeneic NK cell immunotherapy with percutaneous irreversible electroporation enhances the progression-free survival and overall survival in stage III PDAC and extends the overall survival in stage IV PDAC [107,108].

The restoration of NK cell-mediated anti-tumor activity of endogenous NK cells was observed in mouse models in response to chemotherapy. The gemcitabine-mediated anti-tumor effects were in fact dependent on an increase of NK cell infiltration into the tumors and the decrease of myeloid-derived suppressor cells [77]. Independently, it was demonstrated that gemcitabine may also trigger NK cell-mediated tumor cell killing via induction of the surface expression of ligands for the activating receptor NKG2D (NKG2D-L) on tumor cells, while reducing its soluble inhibitory ligands [109] [110,111]. In vitro data revealed that the treatment of PDAC cells with histone deacetylase inhibitors induced the NKG2D-L surface expression rendering the cells more susceptible to NK cell-killing [112]. In general, the administration of NK cell-based immunotherapy against pancreatic cancer has delivered promising results. However, a better understanding and better strategies to overcome the immunosuppressive TME diminishing NK cell activity are needed for further studies. Taken together, there is evidence for an important role of NK cells in PDAC and their potential therapeutic impact, although further research is required to determine the complex interactions between NK cells and the PDAC TME.

## 4. Contribution of Neutrophils in PDAC

The presence of neutrophilic granulocytes (neutrophils) in neoplasms and thus as part of the tumor immune landscape was appropriately described more than 150 years ago as “lymphoreticular infiltration” by Rudolf Virchow. As the most numerous of leukocytes in adult human peripheral blood (up to 70%), neutrophils have a short half-life; however, their abundance suggests that they are an important cell type in the tumor microenvironment of PDAC. Generally, pancreatic cancer cells can recruit polymorphonuclear neutrophils (PMNs) to the tumor vicinity and the desmoplastic stroma, but mostly fail to mount an anti-tumor response. Instead, PMNs are thought to promote tumor progression such that high neutrophil counts correlate with poor patient prognoses. In clinical studies with PDAC patients, a neutrophil-lymphocyte ratio (NLR) was initially assessed as a predictive parameter; however, the diagnostic value of this ratio with regard to prognosis and survival prediction is controversial [112,113,114,115,116,117]. All of these studies are based on the assumption that PMNs can promote pancreatic tumor progression so that high PMN numbers in the tumor microenvironment (TME) correlate with a tumor-progressing state. This view needs to be reconsidered thoroughly.

Initially, as the tumor microenvironment in PDAC is inflammatory in nature, tumor cells secrete pro-inflammatory factors, such as TNF-α and IL-12, that recruit an influx of PMNs into the tumor site. In turn, PMNs secrete a number of chemokines, such as CCL2 (MCP-1), CCL3 (MIP-1α), CCL19, and CCL20 (reviewed in [118]), to attract monocytes and dendritic cells to the TME. As PMNs secure the survival of the host by resolving inflammation, their role in inflammation-driven tumorigenesis is indubitable. PMNs participate in close cross-talk with the TME and consequently, their activation/polarization state changes with the molecular cues with which they are confronted and they constitute a population of tumor-associated neutrophils (TANs). TANs are a major source of pro-inflammatory cytokines, including IL-12, TNF-α, and GM-CSF, as well as chemokines, including CCL-3, CXCL-8, and CXCL-10. These factors affect the migration of cancer cells, as well as other immune cells, in particular, lymphocytes. By specifically releasing IL-10 and TNF-α, TANs reduce the number of lymphocytes and even lead to their dysfunction, such that the outcome of this signaling is a suppressed immunologic reaction at the tumor site. Therefore, a high content of neutrophils in conjunction with a low number of lymphocytes results in a high NLR that is prognostic for a poor patient prognosis. CXCL-10 and CCL-21 released from neutrophils can promote the migration of pancreatic cancer cells toward sensory neurons, thus explaining the generation of pain in PDAC patients [119]. Similar to macrophages, the TME can influence the molecular features of TANs that affect their polarization state. As essential factors, Transforming Growth Factor-ß and Interferon-α (type I IFN) are able to polarize TANs into a TAN N1 and TAN N2 subpopulation, respectively (reviewed in [120]) (Figure 3).

The tumor-associated neutrophil N1 and N2 subpopulations are thought to have different functions in cancer. N1 neutrophils release pro-inflammatory or immunostimulatory cytokines, such as interleukin (IL)-12, tumor necrosis factor (TNF)-α, CCL3, CXCL9, and CXCL10, which facilitates the recruitment and activation of CD8+ T cells [121]. In contrast, N2 neutrophils have strong immunosuppressive and tumor-promoting activity, including the promotion of tumor angiogenesis, invasion, and metastases via various factors, such as hepatocyte growth factor (HGF) [122], oncostatin M [123], reactive oxygen species (ROS) [124], reactive nitrogen species (RNS), matrix metalloproteinase (MMPs), and neutrophil elastase (NE) [118]. In human cancers, the presence of TANs is far more complex. During cancer progression, three distinct populations can be identified based on their densities, with high-density neutrophils (HDN) as mature cells and low-density neutrophils (LDN) containing mature and immature subpopulations. Similar to N1/N2 phenotypes in mice, a dichotomy of neutrophils is maintained as HDNs present cytotoxic capacity towards tumor cells, whereas mature LDNs present suppressive properties usually associated with MDSCs [125].

TANs were shown to localize at the margin of tumor sites in early stage cancer, but they can massively infiltrate into the center of the tumor at later neoplastic stages. The activation of neutrophils, in addition to cytokines and chemokines, is the first molecular event, followed by the recruitment of TANs to tumor tissues. This process is dependent on the presence of CD177 (a neutrophil-specific antigen) that is negatively correlated with the survival of PDAC patients [126]. Among the genes co-expressed with CD177 is phosphodiesterase (PDE) 4D, which is an enzyme involved in multiple tumor-related pathways. In tumor cells, PDE4D upregulates the expression levels of RAS and RAF [126]. In neutrophils, PDE4D causes enhanced phagocytosis, which is linked to increased ROS production. A continuous release of ROS leads to local hypoxia, thereby enhancing neutrophil infiltration [126]. Since local hypoxic changes are known to determine the immune landscape in PDAC, neutrophils are crucially involved in modeling the TME.

An essential feature of human mature neutrophils is the presence of cytoplasmic granules to allow fusion with the plasma membrane (exocytosis, azurophilic granules), endocytic vacuoles (endocytosis, secondary granules), or other granules (tertiary granules). Azurophilic granules play a major role in antimicrobial defense. In contrast, secondary and tertiary granules contain proteins that interact with and degrade the extracellular matrix (ECM). Thereby, tertiary granules are the major constituents in shaping the TME of PDAC and proteins localized to these granules can be highly mobilized. For instance, they contain a high amount of matrix metalloproteinase-9 (MMP-9). When released from these granules, MMP-9 is not in a complex with its physiological inhibitor TIMP-1 [127], so MMP-9 exerts its catalytic activity in the TME, leading to an enhanced invasion of tumor and immune cells by the cleavage of ECM proteins (collagens, laminin, and fibronectin), but also to enhanced tumor angiogenesis by mobilizing Vascular endothelial growth factor (VEGF) from the ECM. With regard to immunosuppression, tertiary granules can release arginase-1 (ARG-1) that induces a down-regulation of CD3ζ and concomitantly, a suppression of CD3-mediated T cell activation and proliferation [128]. A distinct neutrophil population, characterized by a high CD13 content, can cause this type of immunosuppression in PDAC [129].

Neutrophil extracellular traps (NETs) are another unique feature of neutrophils. These are composed of a meshwork of DNA fibers released from neutrophils, together with proteolytic enzymes, to fight against foreign pathogens. However, recent studies suggest that NETs can also contribute to metastasis. The effects of NET inhibitors on spontaneous PDAC mouse models were evaluated: DNase I, which is a NET inhibitor, suppressed liver metastasis [130]. Concurrently, the number of CAFs that accumulated in metastatic foci was significantly decreased. In vitro experiments revealed that pancreatic cancer cells induced NET formation and, consequently, NETs enhanced the migration of hepatic stellate cells that were the possible origin of CAFs in liver metastases. This demonstrates that NET formation in neutrophils can promote liver micrometastasis in PDAC via the activation of CAFs. Recently, a study reported on the effect of IL-17 and identified neutrophil recruitment and NETosis as the major effects of this cytokine. As a result of neutrophil recruitment and NETosis, CD8^+^ T cells were excluded from tumors. As a consequence, IL-17 mediates the resistance of PDAC to checkpoint inhibitors PD1 and CTLA4 [131].

Another role of neutrophils in metastasis is linked to their potency to enter the tissue from the circulatory system. In the bloodstream, neutrophils can escort tumor cells, mostly circulating tumor cells that are disseminated from the primary tumor site [132]. During the extravasation process and after infiltration into metastatic sites, neutrophils undergo apoptosis and NETosis, so that their cargo proteins are released and these create a pro-tumorigenic microenvironment. This includes the release of ARG-1 to create an immune-suppressed environment; MMP-9 to promote the infiltration of tumor cells and tumor angiogenesis; and other potent serine proteases, e.g., neutrophil elastase (NE), Cathepsin G, and human proteinase 3 that can shape the TME at the metastatic site. Other proteins released from neutrophils are lipocalin-2 (NGAL/LCN-2, [133]), heparanase, CD11 and CD18 (LFA-1), and S100/A9. For instance, through the release of lipocalin-2, neutrophils can activate tumor cells via the phosphorylation of PI3K/Akt, causing the downstream expression of VEGF and HIF-1, which are factors important for tumor progression, angiogenesis, and chemoresistance [134].

Given their biological characteristics and short lifespan, there is more evidence for a tumor-promoting than tumor-suppressing effect of neutrophils in PDAC. Due to the instant release of proteins associated with the infiltration of tumor cells and suppression of the immune response, recent data point towards a more active role in the TME, in which neutrophils affect the activation states of other cell types in the immune landscape, such as CAFs and T cells.

## 5. The Role of Adaptive T Cells in PDAC

### 5.1. Accumulation of T Cells

PDAC is characterized by a high immunological heterogeneity, with tumors having variable degrees of T cell infiltration, comprising distinct T-cell subpopulations [14,135,136,137,138,139,140]. Although it has previously been thought that the immunosuppressive microenvironment consisting of fibroblasts and desmoplastic stroma restricts T cell infiltration [141,142], more recent work indicates that desmoplastic elements might not influence T cell accumulation, revealing a specific spatial distribution of T cells in PDAC [139]. The accumulation of CD8^+^ cytotoxic T lymphocytes (CTL) in proximity to cancer cells correlates with increased patient survival [139]. Moreover, the increased presence of CD8^+^ T cells, CTLs, and regulatory T (Treg) cells and decreased numbers of CD4^+^ T cells was detectable in tumors of long-term survivors (LTS) compared to short-term survivors (STS) [140]. In another study, high tumor infiltration by CD4^+^ and CD8^+^ T cells, while low infiltration by Treg cells, and a high ratio of M1 versus M2 macrophages, significantly correlated with longer survival in patients [143]. These studies indicate that the differential effects of T cells in PDAC are dependent on the spatial distribution, type of subpopulation involved, and accompanying macrophage infiltration. Considering that CD4^+^ and CD8^+^ T cells enclose several subpopulations with specific regulation, effector cytokine production, and functions in immunity [144,145], it is not surprising that the T cell effects in PDAC depend on the individual T cell subtype involved (Figure 4).

### 5.2. Th1 and Th2 Responses in PDAC

The subpopulations of CD4^+^ T cells include IFN-γ-producing Th1 cells expressing the master regulator T-BET. Th1, via its marker cytokine IFN-γ and the production of cytotoxic molecules, promotes cellular type I immunity, including the priming, activation, and recruitment of CTLs, M1 macrophages, and NK cells, and mediates immunity against intracellular pathogens and tumors [146]. In contrast, Th2 cells secreting IL-4, IL-5, and IL-13 are characterized by the master transcription factor GATA3 and they coordinate humoral type 2 immunity with the induction of M2 macrophages, which is efficient against helminths and contributes to allergy and asthma. There is a reciprocal relationship between Th1 and Th2 development, in which Th1 inhibits Th2 differentiation and vice versa [145].

A predominant Th2 (GATA3^+^) over Th1 (T-BET^+^) cell infiltration is detectable in pancreatic cancer and an increased ratio of GATA3^+^/T-BET^+^ tumor-infiltrating lymphocytes associated with disease progression, indicating a pathogenic shift towards a Th2 response [147,148,149]. Moreover, higher levels of a Th2 cytokine profile in the serum of patients with resectable pancreatic adenocarcinoma are associated with a shorter overall survival rate [150], thereby confirming the pathogenic role of Th2 cells and their secreted products in PDAC. The Th2 marker cytokine IL-4 promoted the proliferation of human pancreatic cancer cells and enhanced the activation of STAT3, AKT, and MAPK pathways [151], suggesting direct tumor-promoting activity (Figure 4). The induction and accumulation of Th2 cells in the tumor microenvironment were driven by thymic stromal lymphopoietin (TSLP)-activated myeloid dendritic cells (DCs) expressing the receptor for TSLP [147]. In turn, TSLP was secreted by cancer-associated fibroblasts (CAFs) activated by IL-1α and IL-1β, which are products of cancer cells and tumor cell-conditioned macrophages [152]. Basophils expressing IL-4 present in the tumor draining lymph nodes of patients with PDAC probably contribute to the stabilization of the Th2 phenotype, since their presence correlates with an increased Th2/Th1 ratio in tumors and poor patient survival [153]. The Th2-driven responses promoting PDAC also depend on B cells and FcRγ^+^ tumor-associated macrophages (TAMs) expressing Bruton tyrosine kinase (BTK) independent of PI3Kγ. The pharmacological inhibition of BTK or PI3Kγ shifted the immune response towards the Th1 type, accompanied by an activation of CD8^+^ T cells, resulting in PDAC suppression [154]. Moreover, Toll-like receptor signaling and the PDAC microbiome promote Th2-driven immune responses [155,156]. Bacterial depletion slowed tumor growth and germ-free mice were protected from PDAC. This was accompanied by type 1 immune responses with M1 macrophage differentiation and Th1 and CD8^+^ T cell activation [156]. Therefore, Th2 responses can be induced by an interdependent network of tumor cells, CAFs and DCs, by B cells and FcRγ-expressing TAMs activated via BTK and PI3Kγ, as well as by microbiota and associated TLR-signaling. Influencing these tumor microenvironment cellular components via specific pathways leads to a shift towards protective Th1 responses accompanied by the activation of M1 macrophages and CTLs restricting tumor progression.

### 5.3. Th17 and Treg Responses in PDAC

Besides Th1 and Th2 cells, Th17 and Treg cells are also involved in the regulation of PDAC. Th17 cells produce IL-17, IL-21, and IL-22, depending on the lineage-specific transcription factor RORγt, and have a fundamental function in protection from extracellular bacterial and fungal infections. In contrast, Foxp3^+^ Treg cells have anti-inflammatory properties and display plasticity towards Th17 cells [145,157]. In PDAC, increased Th17 frequencies were detected in tumors compared to adjacent normal tissue, whereby advanced tumors (stage III-IV) harbored significantly higher Th17 cell proportions compared to stage I-II tumors. Furthermore, a higher presence of IL-17^+^ T cells in tumor tissue correlated with a shorter overall survival and the occurrence of metastases. Finally, the systemic levels of IL-17 were increased in PDAC patients compared to healthy donors and correlated with the pancreatic cancer severity [158,159]. These reports indicate an involvement of IL-17 and Th17 cells in PDAC progression.

IL-17 contributes to the initiation of carcinogenesis since the neutralization of IL-17 prevented early neoplastic lesion formation (pancreatic intraepithelial neoplasia (PanIN), which can progress to invasive ductal adenocarcinoma), while forced IL-17 expression accelerated PanIN formation and progression. The recruitment of Th17 cells into the tumor microenvironment is induced by oncogenic Kras expressed in early pancreatic neoplastic epithelium, which mediates the expression of the receptor for IL-17 in a cell autonomous manner [160] (Figure 4). In this study, IL-17 enhanced early PanIN formation via the induction of REG3β, and deletion of this gene decreased PanIN lesions in oncogenic Kras-driven pancreatic cancer initiation. Additionally, REG3β promoted cell growth and survival through activation of the gp130-JAK2-STAT3 pathway [161]. Besides these effects, IL-17 regulated PDAC stem cell features via the upregulation of DCLK1, POU2F3, and ALDH1A1, and the expression of the IL-17A/F-specific receptor—IL17RC. Along with this finding in the mouse model, in human pancreatic cancer tissue, high DCLK1 or POU2F3 levels are associated with a shorter median survival time of patients [162]. This indicates multiple direct IL-17 effects on pancreatic cancer initiation, including the regulation of growth, PanIN formation, and stem cell properties via the regulation of REG3β, DCLK1, POUF3, ALDH1A1, and IL-17RC.

Mechanistically, the alternative pathway of p38 mitogen-activated protein kinase (MAPK) activation (via the phosphorylation of Tyr323 (pY323) on p38) contributes to the induction of Th17 cells in PDAC and a high proportion of tumor-infiltrating Th17 cells displayed p38-pY323 positivity, which correlated with aggressive disease in humans. Moreover, in the mouse model, genetic ablation of the alternative p38 pathway inhibited the growth of tumors. In turn, the transfer of p38-proficient, but not of p38-deficient, T cells enhanced tumor growth [163]. Additionally, PD-L1 expressed by T cells promoted Th17 polarization [164]. Besides IL-17, IL-21 which is produced by Th17 cells, enhances pancreatic cancer invasion via induction of the transcription factor BLIMP-1. The accumulation of IL-21^+^ T cells, as well as BLIMP-1 and IL-21R positivity by tumor cells, is associated with a poor patient survival [165].

The master-regulator of type 17 T cell differentiation—RORγ—is coopted by pancreatic cancer cells and promotes their aggressive phenotype (Figure 4). RORγ expression is upregulated during pancreatic cancer progression and, consequently, its inhibition restricted growth and improved survival in a mouse model of pancreatic cancer [166]. In patients, its expression correlated with advanced disease and metastasis [65], indicating that the acquisition of RORγ-directed programs, by both pancreatic cancer and T cells, has a tumor-promoting intrinsic or extrinsic consequence.

### 5.4. Regulatory T Cells (Tregs) and PDAC

There are several studies indicating the tumor-promoting roles of regulatory T-cells. The prevalence of Tregs was increased in the peripheral blood, as well as in the tumor microenvironment, of patients with pancreatic cancer—a finding not only limited to the phase of disease progression, but also as early as in premalignant lesions [167,168]. Accordingly, Tregs are detectable in precursor lesions in intraductal papillary mucinous neoplasm (IPMNs) and with disease progression, there is a marked increase in the Treg/CD8+ T cell ratio [169]. Furthermore, a low abundance of Tregs is associated with a better prognosis and the conversely increased prevalence of beneficial CTLs [168]. Accordingly, tumor infiltration with low numbers of Tregs and high numbers of M1 and CD8^+^ T-cells significantly correlated with longer survival [143], indicating an inverse relationship between beneficial type 1 responses, including M1 macrophages and CD8^+^ T-cells, and rather detrimental Treg cells. Consistently, in the mouse model, the proportion of Treg cells increased during tumor progression and genetic Treg ablation after the onset of tumors slowed down tumor growth and prolonged overall animal survival [170]. Mechanistically, Tregs induced a tolerogenic phenotype (CD11c^+^ DCs) (Figure 4) characterized by a decreased expression of MHC class II, CD40 and CD86 co-stimulatory molecules, and indoleamine 2,3-dioxygenase (IDO), which suppressed beneficial IFN-γ production and the activation of CTLs [170]. Treg plasticity towards Th17 cells probably also contributes to PDAC progression, since TGFβ and IL-17A produced by Tregs were significantly associated with a poor prognosis and TGFβ, IL-17A, and IL-6 levels were significantly lower in patients responding to chemotherapy than in non-responders [159]. Moreover, PDAC patients were shown to possess elevated numbers of pro-inflammatory and immunosuppressive FOXP3^+^RORγt^+^ Tregs producing Th17 and Th2 cytokines [171].

Contrary to this data, a recent report revealed an increased intra-tumoral presence of Treg cells and CD8^+^ T cells (CTLs), while decreased numbers of CD4^+^ T cells were found in tumors of long-term survivors (LTS) compared to short-term PDAC survivors (STS) [140]. Along with this finding in patients, the genetic depletion of Tregs during the onset and progression of murine pancreatic cancer caused disease acceleration by modulating the cellular components of the tumor microenvironment [172]. The changes included the induction of tumor-promoting inflammatory cancer-associated fibroblasts (iCAFs) instead of tumor-restraining myofibroblastic CAFs (myCAFs), which were dependent on TGFβ secreted by Tregs. Moreover, Treg depletion induced the expression of CCL3, CCL6, and CCL8 chemokines by epithelial cells and fibroblasts, which recruited myeloid cells in a CCR1-dependent manner, thereby promoting immune suppression [172] (Figure 4). Taken together, Tregs display multifaceted roles in pancreatic cancer dependent on the tumor stage and environment. On one hand, they can promote tumorigenesis by suppressing T cell activity via immunomodulatory effects on DCs [170] or by plasticity towards Th17 cells [159,171]. On the other hand, they can restrain tumor growth by promoting myCAF formation and balancing the suppressive myeloid cell compartment [172].

### 5.5. CD8^+^ T Cell Responses in PDAC

The main subpopulation of CD8^+^ T cells, namely the CTLs producing IFN-γ, TNF, and cytotoxic molecules including perforin and granzymes, are the best effectors in tumor rejection because of their specificity and cytotoxic activity against tumor cells. They form a long-term existing memory, thereby establishing protection from cancer recurrence [149]. However, tumors induce a dysfunctional state in CTLs termed *exhaustion*, which is characterized by an impairment of effector function, reduced survival, and increased expression of inhibitory receptors including the checkpoint inhibitors programmed cell death 1 (PD1) and cytotoxic T-lymphocyte-associated protein 4 (CTLA-4) [173,174]. Immunotherapy of tumors by a checkpoint blockade has led to durable responses in melanoma, non-small cell lung carcinoma, head-and-neck cancer, and urothelial carcinoma [174]. In pancreatic cancer, an increased accumulation of CD8^+^ T cells in tumor tissue and close proximity to tumor cells correlate with a better survival [139,140,143]; however, many CD8^+^ T cells express checkpoint inhibitors, including PD1 and other inhibitory receptors, and in tumors, PD-ligand 1 (PD-L1) is detectable [136,138] (Figure 4). In addition to PD1, T cells can also express PD-L1, which equally suppresses CD8^+^ T cell responses to PD1 and promotes M2-like macrophages [164].

The application of checkpoint inhibitors as monotherapy has been unsuccessful in PDAC, prompting the introduction of combination therapies and a better stratification of patients [175,176]. Therefore, understanding CD8^+^ T cell driving factors, including neoantigen generation, as well as mechanisms governing the CD8^+^ T cell infiltration of tumors, will be crucial for the development of further therapy regimens. An establishment of successful immunity in long-term PDAC survivors required a high neoantigen quantity, together with a high abundance of intra-tumoral CD8^+^ T cells [140]. Besides the quantity, the quality of neoantigens was established as a biomarker of survival in PDAC. Potent neoantigens were targeted by cross-reactive T cells recognizing both cancer neoantigens and homologous non-cancer microbial antigens. Among these neoantigens, MUC16, which is a common ovarian cancer biomarker, was identified as a candidate immunogenic hotspot in PDAC. Therefore, neoantigens with a high quality are crucial for an efficient CD8^+^ T cell response in immunogenic pancreatic tumors [140].

Tumor cell-intrinsic factors seem to shape T cell infiltration, survival, and response to the treatment [138]. In a mouse, CD8^+^ T cells in both T-cell inflamed and T-cell non-inflamed tumors stemming from congenic tumor cell clones displayed a low functional status and expressed similar exhaustion markers (PD1, CTLA-4, and TIM3). However, in contrast to T-cell non-inflamed tumors, T-cell inflamed tumors were rejected after immunotherapy consisting of an anti-CD40 agonist, anti-CTLA-4, and anti-PD1. Furthermore, cured mice developed memory to both inflamed and non-inflamed tumor cells. This highlights the importance of a high abundance of pre-existing intratumoral PD1^+^CD8^+^ T cells for a successful and long-lasting response to immunotherapy and sensitivity of non-inflamed tumors to shared antigen recognition by T cells. The T-cell infiltration and sensitivity to immunotherapy were governed by cross-presenting CD103^+^CD11c^+^ DCs, which were associated with inflamed tumors. In contrast, non-inflamed tumors attracted CXCR2^+^ myeloid and myeloid-derived suppressor cells (MDSC) via CXCL1. This prevented CD8^+^ T cell infiltration, thereby revealing that tumor-cell-intrinsic factors shape the interdependent immune cell migration and cross-talk, allowing responsiveness to immune therapy [135]. This is consistent with a previous report showing that myeloid cells support PDAC and influence CD8^+^ T cell infiltration [69]. Therefore, besides neoantigen generation, understanding the mechanisms governing the infiltration of tumors by CD8^+^ T cells will be important for the design of future combination immunotherapies, in order to improve the treatment success of PDAC patients.

## 6. Cancer-Associated Fibroblasts (CAFs) as Immunological Modulators in PDAC

Strictly speaking, stellate cells, fibroblasts, and cancer-associated fibroblasts (CAFs) are not of hematopoietic origin and do not belong to the group of immune cells. Nevertheless, given the highly desmoplastic nature of PDAC and the high abundance of CAFs in this disease, these cells are central players in the tumor microenvironment, serving as signaling hubs and interacting with most, if not all, cell types within a pancreatic tumor, including immune cells. Hence, it appears justified, if not mandatory, to include fibroblastic cells in this review as prominent modulators of the immune cell compartment.

CAFs can functionally interact with immune cells in various ways, including through mechanical cues, direct cell-cell contact, metabolic cross-talk, or the secretion of soluble immune mediators. The mode of interaction is in part dictated by the type of CAF being investigated. Single cell transcriptomics has recently identified at least three different CAF subpopulations in PDAC with different spatial distribution and functional properties [177,178,179]. CAFs closest to cancer cells display properties of activated fibroblasts, show contractility, and are therefore considered to be myofibroblastic CAFs (or myCAFs). This cellular subtype responds to the high local levels of tumor cell-secreted TGFβ, which drives the expression of, e.g., *ACTA2* (Smooth muscle actin, *SMA*) and *Collagen* genes (Figure 5A). In addition, a second CAF subtype was identified, which possesses immune-modulating properties and which, when compared to myCAFs, is located more distantly from cancer cells. These inflammatory-CAFs (or iCAFs) are characterized by marker proteins such as IL-6 and other inflammatory mediators, such as IL1, IL21, LIF, and CXCL1-3. The transdifferentiation of CAFs or quiescent pancreatic stellate cells (PSCs) into iCAFs is mediated by IL1-induced JAK/STAT signaling. The spatial segregation of the two CAF subtypes is mechanistically brought about by TGFβ signaling downregulating the IL1 receptor, essentially rendering these cells unresponsive to IL1 ligands [180]. An inflammatory CAF population secreting IL6 and CXCL12 and a separate *ACTA2^+^* myCAF population were also described by other independent studies [181,182] (Figure 5B).

Recently, a third CAF subtype was described by Tuveson and colleagues [177,179], which might represent a separate subset within the iCAF population, but expressed characteristic marker molecules, such as *MHCII*, *CD74,* and *SLPI*. Hence, these cells were termed antigen-presenting CAFs (or apCAFs). ApCAFs were able to present the ovalbumin model antigen (OVA) in an MHCII-dependent manner and activate CD4^+^ T-cells [179]. However, as they lack the expression of co-stimulatory molecules such as *CD80* or *CD86*, it currently remains unclear whether they might induce T-cell anergy in vivo or not. As a result, apCAFs might potentially represent yet another immune-suppressive cell population within the TME of PDAC.

Importantly, these two (or three) CAF subtypes are not static, but very dynamic, and are instructed by local signaling networks, making them very adaptive, but also amenable, to therapeutic approaches [180]. Although the number of available studies is limited at this point, it is generally assumed that myCAFs exert tumor-restraining functions, whereas iCAFs rather possess tumor-promoting roles. As such, myCAFs could readily be identified in early PDAC precursor lesions, whereas iCAFs were frequently found in late-stage PDAC [181]. Hence, for therapeutic approaches, the reprogramming of CAF subtypes might be more promising than broadly and non-specifically depleting a tumor of all kinds of CAFs. In line with this hypothesis, the experimental depletion of CAFs in PDAC mouse models resulted in aggravated disease with increased angiogenesis and prominent infiltration of immunosuppressive Tregs [183,184]. When discussing the contribution of CAFs to the immune cell compartment, iCAFs are likely the most important regulators to focus on. Nevertheless, myCAFs might still contribute to the tumoral immune landscape by the synthesis of collagens and other ECM components, which results in tissue stiffness and an increased interstitial fluid pressure, potentially hindering immune cell infiltration [185,186]. However, the ECM is not a simple barrier of cell invasion [139] and the situation seems more complex than initially anticipated. In the skin, for example, fiber alignment and the type of crosslinking of the fibroblast-generated extracellular matrix (ECM) have been shown to selectively impair the migration of CD8^+^ T cells, while supporting the infiltration of Tregs and MDSCs [187], suggesting that myCAFs and the ECM might also play significant roles in PDAC and that targeting the ECM might be of clinical use. The best understood myCAF inducer is TGFβ, which upregulates the protein βig-h3 in stromal cells and which directly suppresses CD8^+^ T-cell activity and promotes TAM M2 polarization [188]. Another component of the acellular TME is hyaluronic acid (HA), which is the target of PEGPH20—a formulation of hyaluronidase (the HA-degrading enzyme). Several clinical trials with PEGPH20, in combination with standard of care, but also with immune checkpoint inhibitors, are ongoing.

ICAF formation can be triggered by the TLR4-mediated induction of IL-1β in tumor cells, which promotes the transdifferentiation of CAFs into iCAFs [189]. Intriguingly, bacteria (intra-tumoral or gut commensal) might contribute to or elicit stimulation of the TLR4 receptor. The resulting iCAF population was shown to have an impressively widespread impact on the immune landscape of the experimental tumors, promoting not only M2-polarization of tumor-associated macrophages (TAMs), but also an increase in intra-tumoral numbers of myeloid-derived suppressor cells (MDSCs), tumor-associated neutrophils (TANs), regulatory B-cells (Bregs), and Th17 cells, as well as a decrease in CD8^+^ cytotoxic T-cells [189]. In summary, this exemplifies the importance of CAFs as potent modulators of the immune compartment in PDAC.

ICAFs are a rich source of numerous soluble immune modulators, such as IL-6, IL-8, IL-33, LIF, and CXCL12, to name a few. All of these play critical roles in pancreatic cancer development. Besides the pro-proliferative and pro-survival effects of interleukin-6 (IL-6) on tumor cells, this inflammatory mediator is well known for its involvement in regulating a vast array of immune cells. For instance, IL-6 and its downstream effector STAT3 induce the differentiation of myeloid cells into immunosuppressive MDSCs or M2 TAMs. Furthermore, it negatively regulates neutrophils, natural killer cells, effector T cells, and dendritic cells (reviewed in [190]).

In addition, CAFs secrete IL8, which positively impinges on macrophage growth and migration [191,192]. Pancreatic CAFs have also been shown to secrete high amounts of IL-33, which polarizes TAMs towards a tumor-promoting M2 phenotype [193]. However, it should be mentioned that the situation with regard to the pro-tumorigenic function of IL-33 is less clear as this cytokine also promotes group 2 innate lymphoid cells (ILC2s), which act as immune activators in pancreatic cancer [194]. Nevertheless, with respect to bone marrow-derived monocytes, it was shown that they are recruited to tumors by CAF-derived CCL2, where they can differentiate into M2-TAMs [48]. CCL2 also promotes tumor angiogenesis at metastatic sites, suggesting that the targeting of this chemokine would be a promising strategy [195]. However, a recent phase Ib trial investigating the inhibition of the CCL2 receptor (CCR2) in combination with nab-paclitaxel/gemcitabine did not show an improvement compared to chemotherapy alone (NCT02732938; [196]).

CAFs are also a considerable source of CXCL12, which impacts cancer cells in a pro-migratory manner, but also renders them resistant to T cell-mediated killing. Stroma-derived CXCL12 also suppresses cytotoxic T-cell infiltration [197]. In turn, blocking CXCR4 (the receptor for CXCL12) promoted T-cell infiltration and synergized with immune checkpoint approaches [142,198]. Clinical trials (e.g., NCT03386721) targeting the CXCL12-producing FAP^+^ stromal subpopulation are ongoing [199].

An important member of the CAF secretome in PDAC seems to be the leukemia inhibitory factor (LIF). LIF can have direct effects on cancer cells, determining, for instance, the degree of cell differentiation, epithelial-mesenchymal transition (EMT) status, chemoresistance, and metastasis [200,201,202]. On the other hand, LIF also controls the immune tumor microenvironment by hindering CD8^+^ T cell tumor infiltration and promoting the presence of pro-tumoral macrophages [203]. Interestingly, this reflects the role of LIF in early embryonic development, where LIF generates a local immunosuppressive microenvironment in the uterus in order to protect the embryo from the mother’s immune system [204].

Although a major source of TGFβ ligands is represented by the tumor cell compartment, activated fibroblasts have also been described as producers of this broadly immunosuppressive mediator in several instances [205,206,207]. Furthermore, regulatory T-cells express TGFβ in the immune TME, where it promotes immune evasion by dampening the functions of antigen-presenting cells (APCs) and effector T-cells [172,208,209]. As such, Tregs can impinge on the TGFβ-triggered transdifferentiation of CAFs into myCAFs [172].

Furthermore, immune cells might be compromised in their functionality by metabolic competition with the abundant stromal fibroblast population. The latter cells can undergo the so-called *Reverse Warburg* effect, essentially taking up extracellular glucose and metabolizing it to lactate, which is in turn provided to cancer cells as anabolic building blocks. This would create glucose scarcity in the TME, thereby negatively affecting the T-cell effector capacity—a process which itself requires sufficient glycolytic activity for correct functioning [210].

In addition to the aforementioned metabolic and soluble mediators utilized by mesenchymal cells to cross-talk with immune cells, CAFs can also suppress the cytotoxic capability of tumor-infiltrating CD8^+^ T cells in a direct cell-to-cell manner [211]. Specifically, CAFs possess antigen cross-presenting activity via MHCI, resulting in the engagement of CD8^+^ T-cells and the subsequent upregulation of PD-L2 and FASL on CAFs, thereby inactivating and killing intra-tumoral CTLs. Therapeutically blocking the PD-L2/PD-1 interaction might reactivate the cytotoxic T-cell function in this context [211].

Taken together, the highly abundant CAF population functions as a central cellular network regulating important immune cell types in PDAC by indirect ECM and metabolism cross-talk, as well as through soluble mediators and direct cell-cell contacts.

## 7. Immunotherapy in PDAC

The intricate relationship between different immune cell populations has been used to develop immune-based therapies for pancreatic cancer (summarized in Figure 6). Most prominently, checkpoint inhibitors were successful in the treatment of many solid tumor entities, such as melanoma [212]. Checkpoint inhibitors aim to reverse intra-tumoral T cell dysfunction by blocking receptors that are overexpressed during T cell exhaustion, such as PD-1 and cytotoxic T lymphocyte antigen 4 (CTLA-4) [213]. The binding of programmed death-ligand 1 (PD-L1) or PD-L2 to PD-1 leads to ectodomain competition, leaving activating ligands without their respective costimulatory receptors, ultimately resulting in the suppression of T cell effector functions. PD-1 activation has been shown to suppress T-cell motility [214], antagonize T-cell receptor (TCR) signaling [215], and suppress effector gene transcription [216]. In PDAC, the expression of PD-L1 inversely correlated with survival, and anti-PD-L1, as well as anti-PD-1, treatment significantly reduced tumor growth in mice subcutaneously injected with a murine PDAC cell line [217]. However, checkpoint inhibitor treatment alone was ineffective in a genetically engineered PDAC mouse model [218]. In clinical trials, checkpoint inhibitors did not meet the expectations. The anti-CTLA-4 antibody Ipilimumab was tested in 27 patients with advanced pancreatic carcinoma, without beneficial efficacy [219]. Similarly, anti-PD-L1 monotherapy was not effective in pancreatic carcinoma patients [220].

Accordingly, the clinical use of checkpoint inhibition as a monotherapy is currently limited to a very narrow subset of pancreatic carcinoma patients that exhibit mismatch repair (MMR) deficiency. In 2015, Le et al. reported, in a phase II study, that the MMR status predicts the clinical benefit of an immune checkpoint blockade with the anti-PD-L1 drug pembrolizumab in solid cancers (none of them were pancreatic cancer) [221]. A beneficial efficacy was reported for the response rate, progression-free survival, and overall survival. Whole-exome sequencing found a distinctly higher number of somatic mutations per tumor in MMR-deficient samples (mean of 1782) versus MMR-proficient tumors (mean of 73). A high somatic mutational load was associated with prolonged progression-free survival. In 2017, the authors published an expanded version of their analysis, reporting about 12 different tumor types, including pancreatic carcinoma [17]. All six patients with MMR-deficient pancreatic carcinoma had responded to treatment. The study led to accelerated FDA approval of the anti-PD-L1 drug pembrolizumab as a second-line agent for patients with advanced microsatellite instability-high (MSI-H) and MMR-deficient solid tumors, regardless of the histotype. MMR deficiency results in a high mutation rate, due to the failure of the DNA repair mechanisms. This increases immunogenicity, as somatic mutations encode non-self, immunogenic neo-antigens. As a result, tumors with a large number of somatic mutations are more susceptible to a checkpoint blockade. A higher immune cell infiltration of tumors with MMR deficiency had been reported before [222]; however, whole exome sequencing now provides causal evidence that MMR deficiency, somatic hypermutation, and the response to checkpoint inhibitor therapy are interconnected [223].

Unfortunately, however, only one percent of pancreatic carcinoma patients have a defect in their MMR system [224], and the majority of pancreatic carcinoma patients will not benefit from monotherapy with checkpoint inhibitors. Therefore, combination therapies have been suggested as a way to overcome the ineffectiveness of checkpoint inhibitor therapy. This strategy aims to increase the immunogenicity of pancreatic carcinoma. Different from antigenicity, which refers to the mutational burden of malignant neoplasms, as discussed above, immunogenicity refers to the highly immunosuppressive tumor microenvironment that shields pancreatic carcinoma cells from immune cell effector functions [225]. The most direct way to combine checkpoint inhibition is through chemotherapy, as treatment with gemcitabine, nab-paclitaxel, or the FOLFIRINOX regime is standard of care in pancreatic carcinoma. Chemotherapeutic agents ignite intra-tumoral immune responses by inducing immunogenic cell death (ICD). This form of cell death is characterized by a necrolytic release of danger signals, which in turn will change the cytokine milieu, alter the presence of suppressive cell types such as MDSC and Treg, increase the surface expression of MHCI on cancer cells, and influence the maturation of intra-tumoral dendritic cells [226]. In murine models, this approach has resulted in an increased efficacy of checkpoint inhibition. Winograd et al. reported a triple combination of gemcitabine/nab-paclitaxel with anti-PD-1 and immunostimulatory anti-CD40 mAbs using a genetically engineered mouse model of PDAC, resulting in an improved median overall survival of tumor-bearing mice [218]. Another group found that gemcitabine combined with a PD-L1 blockade resulted in a synergistic antitumor effect and complete response in treated mice [217].

Multiple clinical trials have tested the combination of checkpoint inhibition and chemotherapy in PDAC patients [175]. Published studies have reported on the anti-CTLA-4 antibodies tremelimumab and ipilimumab, respectively [227,228], as well as the anti-PD-1 antibody pembrolizumab [229]. However, most of these trials are phase I studies, and no conclusive beneficial evidence has been reported [176]. It has been shown that immunogenic cell death can also be induced by radiotherapy, which has led to strategies aimed at the re-introduction of radiotherapy into the treatment of PDAC. The beneficial combined efficacy in murine models is promising [230,231]; however, to date, results of clinical trials on the combination of radiotherapy and checkpoint inhibition have not been published.

In addition, a multitude of immunostimulatory agents and methods have been tested for their potential to overcome the low immunogenicity of pancreatic carcinoma and induce synergistic efficacy with checkpoint inhibition. Current reviews inform on monoclonal antibody-based therapies such as agonistic anti-CD40 therapy, on vaccination strategies, and on adoptive cell therapies (ACT)—as monotherapy or in combination with checkpoint inhibition [232,233]. ACT therapy expands autologous or allogeneic immune cells ex vivo before reinfusion, avoiding inhibitory signals from suppressive cell populations of the tumor microenvironment [234]. The efficacy of this approach was demonstrated in a murine model of pancreatic carcinoma decades ago [235]. More recently, clinical trials using MUC-1 as a PDAC-associated target reported promising results [236,237].

In a similar approach, T cells, which are genetically modified to express a chimeric antigen receptor (CAR), are adoptively transferred (CAR-T cell therapy). Here, the extracellular CAR domain consists of a single-chain variable fragment (scFv) of an antibody that recognizes a specific tumor antigen. The intracellular domain, however, contains the T-cell receptor signal transduction sequence. CAR-transfected T cells can target any extracellular molecular structure that can be recognized by an antibody, thus abrogating the requirement for MHC restriction. Excellent reviews inform on techniques to manufacture next-generation CAR-T cells [238], and on current CAR-T cell trials in the field of PDAC therapy [239].

## 8. Conclusions

Pancreatic cancer has been considered poorly immunogenic for decades. However, this characterization might be over-simplistic. Recent data have resulted in a refined topography of the immune landscape that defines the pancreatic cancer microenvironment. It has become clear that complex mechanisms regulate the interplay of myeloid, lymphoid, and stromal cellular compartments, resulting in the low immunogenicity and antigenicity of PDAC. However, these complex mechanisms can be overcome by equally sophisticated treatment strategies. From humble beginnings, the first success stories can be reported. Prominently, the efficacy of checkpoint-inhibitor therapy in a small subgroup of PDAC patients with MMR-deficiency has initiated the era of personal medicine for pancreatic carcinoma patients and introduced the first FDA-approved treatment targeting a cellular stromal component. The future direction of pancreatic cancer immunotherapy will depend not only on the outcome of ongoing combinatorial trials in the field of checkpoint blockades, but even more on the ever-increasing understanding of pancreatic cancer’s basic immunology. We envision that, in the coming years, a small number of the currently promising pre-clinical immune-bolstering approaches might find their way into various clinical trials, most likely as combination therapy with chemotherapy or anti-proliferative therapy (e.g., MEK inhibitors). We believe that, in addition to advances in personalized medicine to define responsive patient cohorts, out of the numerous experimental candidates identified so far, selected immune-focusing therapies will eventually make a step forward towards an improved treatment of this devastating disease.

## Figures and Tables

**Figure 1 ijms-21-07307-f001:**
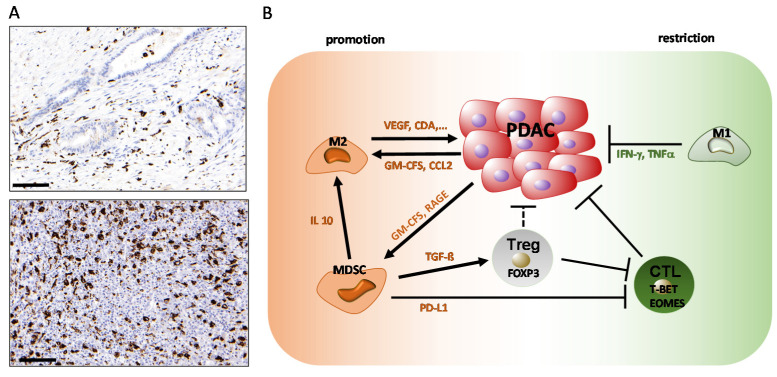
Tumor-associated macrophages (TAMs) and myeloid-derived suppressor cells (MDSCs) in pancreatic ductal adenocarcinoma (PDAC). (**A**) Immunohistochemical staining of pan-monocytic marker CD68 (macrosialin; dark brown signals) in examples of well-differentiated, stroma-rich (upper panel) and poorly differentiated, stroma-poor (lower panel) pancreatic ductal adenocarcinoma. Scale bar: 100 µm. (**B**) Scheme of TAM functions in PDAC.

**Figure 2 ijms-21-07307-f002:**
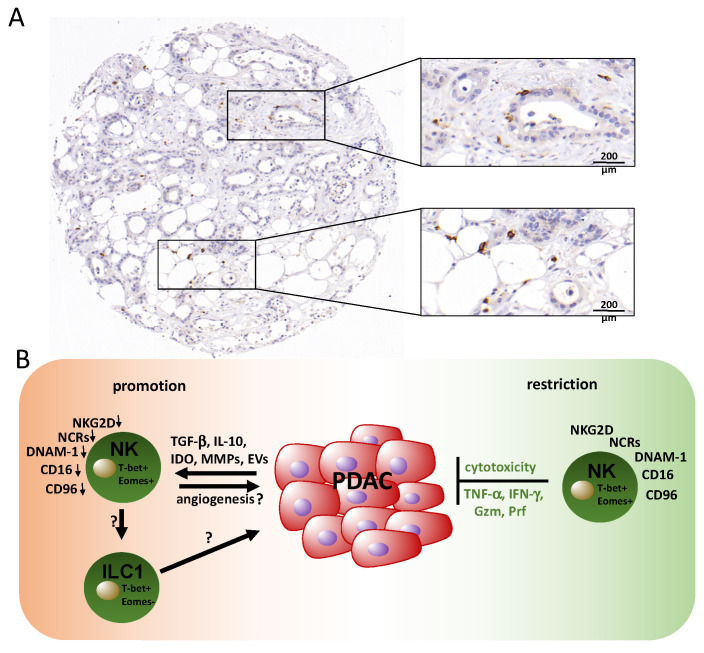
Natural killer cells and their roles in PDAC. (**A**) Immunohistochemical staining of CD56 reveals disseminated natural killer (NK)/NKT (Natural killer T-cells) cells (brown). The depicted case shows a rather high density of NK cells considering the distinct heterogeneity of NK cell infiltration, which is generally low in PDAC tissue. (**B**) PDAC NK cells isolated from the periphery are characterized by a reduced expression of cytotoxicity receptors and exhibit impaired anti-tumor activity, which is induced by mediators of the tumor microenvironment. Restoration of NK cell functions is possible, e.g., via ex vivo activation with IL-2 and gamma-irradiated feeder cell lines. The phenotype of tumor-infiltrating NK cells or other innate lymphocytes such as ILC1 cells and their activity is poorly understood. Gzm: granzyme; Prf: perforin.

**Figure 3 ijms-21-07307-f003:**
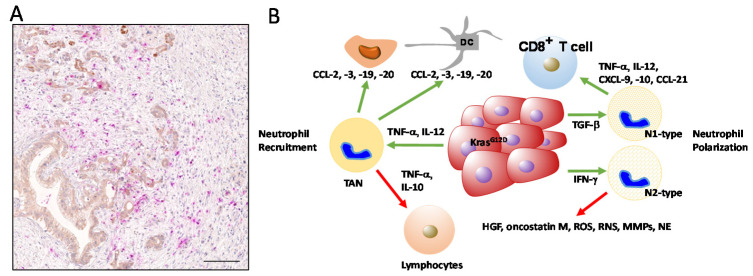
Tumor-associated neutrophils in pancreatic cancer. (**A**) Neutrophils in PDAC tumor stroma, stained with an antibody against Myeloperoxidase (MPO, pink). Ductal structures with tumor cells are stained in brown. Scale bar: 200 µm. (**B**) Signaling events leading to neutrophil recruitment (left side) and neutrophil polarization (right side). Activating signals are indicated by green, and suppressing signals by red arrows. Note that neutrophils can activate macrophages and dendritic cells, while they suppress lymphocytes. When polarized, the tumor-associated neutrophil (TAN) N1 type can activate CD8-positive T cells, whereas TAN N2 secretes a cocktail of tumor-promoting factors, such as hepatocyte growth factor (HGF), matrix metalloproteinase (MMPs), and neutrophils specific protease neutrophil elastase (NE). Furthermore, TAN N2 cells can cause an increased production of reactive oxygen and nitrogen species (ROS and RNS, respectively) and can cause hypoxic signaling.

**Figure 4 ijms-21-07307-f004:**
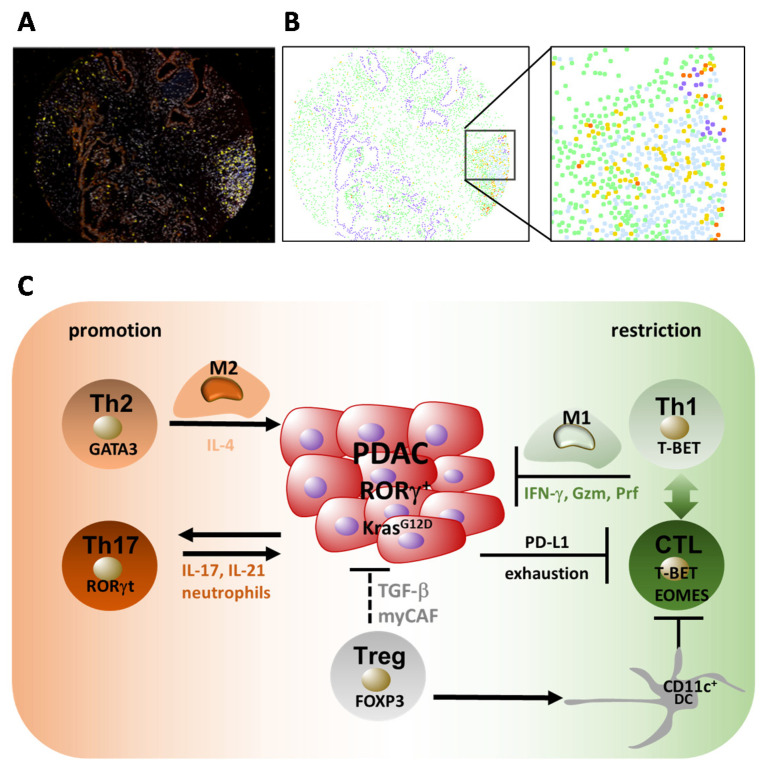
Role of T cell subpopulations in PDAC. (**A**) Compound image of a pancreatic cancer tissue microarray (TMA) section (blue: CD3; yellow: CD8; red: programmed cell death 1 (PD-1); orange: PanCK). (**B**) Phenotype map, same section as in A (yellow: CD8^+^ PD-1^-^ T cells; orange: CD8^+^ PD-1^+^ T cells; purple: epithelium/tumor; light blue: CD8^-^ T cells; green: everything else). Magnified area is densely infiltrated by PD-1^neg^ CD8^+^ T cells (CTLs), as well as PD-1^pos^ CD8^+^ CTLs. Remnants of epithelial/tumor cells are found in close vicinity to exhausted PD-1^pos^ T cells. (**C**) Th2 cells expressing the transcription factor GATA3 can be induced by (i) interleukin (IL)-1α and IL-1β primed cancer-associated fibroblasts (CAFs) producing thymic stromal lymphopoietin (TSLP), which diverts dendritic cells (DCs) towards the Th2 inducing phenotype; (ii) B cells and tumor-associated macrophages in a Bruton tyrosine kinase (BTK)- and Pi3Kg-dependent manner; and (iii) by the microbiome via TLR-signaling. Th2 cells recruit tumor-promoting macrophages (M2) and can directly induce tumor cell proliferation via IL-4. Oncogenic Kras mediates Th17 cell recruitment into the tumor microenvironment. Via secreted IL-17 and IL-21, Th17 cells enhance early PDAC progression. An alternative p38 mitogen-activated protein kinase (MAPK) activation pathway in CD4^+^ T cells directs Th17 cell differentiation. Beneficial CTL and Th1 responses in PDAC are suppressed by Tregs via the immunomodulation of CD11c^+^ DCs. Tregs can be beneficial in PDAC as they suppress tumor progression via the induction of myofibroblastic CAF (myCAFs) by secreted transforming growth factor beta (TGFβ). CTL and Th1 cells restrict PDAC growth via secreted IFN-γ and direct cytotoxicity. Th1 promotes CTL responses and M1 differentiation, thereby restricting tumor growth. CTL accumulation in the tumor microenvironment can be induced by cross-presenting CD103^+^CD11c^+^ DCs, while tumor cells, via CXCL1-dependent attracting myeloid cells, restrict CD8^+^ T cell accumulation in tumors. High-quality neoantigens, including MUC16, contribute to long-term survival in PDAC.

**Figure 5 ijms-21-07307-f005:**
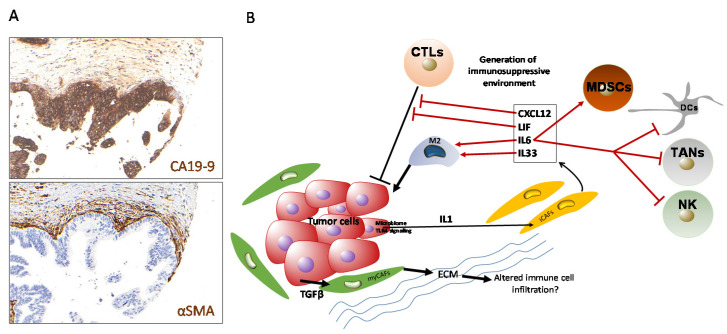
The functions of CAFs in the immune landscape of PDAC. (**A**) Immunohistochemical example of tumor cell localization (CA19-9 staining in brown, upper panel) and αSMA-positive myCAFs (also brown, lower panel) in human PDAC. Note that the myCAFs are in close vicinity to the tumor cells. (**B**) Tumor cells polarize quiescent pancreatic stellate cells towards different subtypes of cancer-associated fibroblasts (CAFs). Contractile myCAFs and inflammatory-CAFs (iCAFs) represent the two major populations. IL1-induced iCAFs secrete a large number of soluble mediators that affect a plethora of immune cells, essentially generating an immunosuppressive environment favoring tumor cell growth. In addition, immune cells might also be suppressed by CAF-mediated metabolic competition and direct cell-cell contact (not shown).

**Figure 6 ijms-21-07307-f006:**
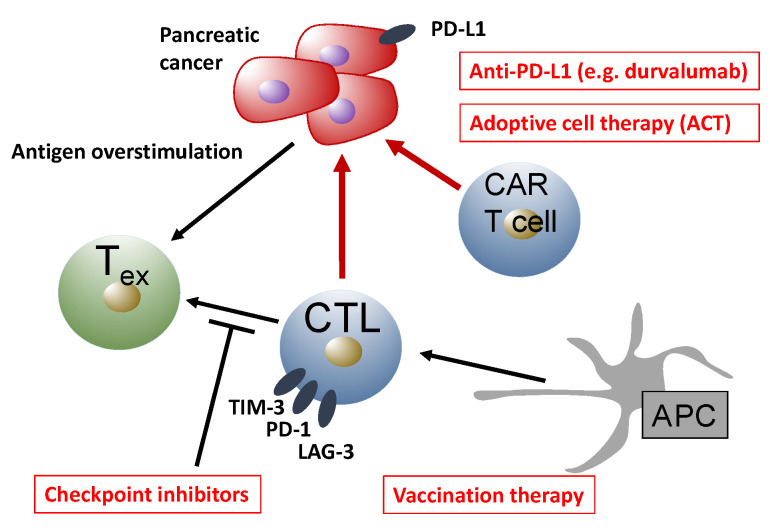
Cross-talk between cancer cells and immune cells mediating immunotherapeutic approaches in pancreatic cancer patients. APC, antigen-presenting cell; CAR, chimeric antigen receptor; CTL, cytotoxic T cell; LAG-3, lymphocyte-activation gene-3; PD-1, programmed death receptor-1; PD-L1, programmed death-ligand 1; T_ex_, exhausted T cell.

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
