# Peer review of "The Immune Microenvironment in Pancreatic Cancer"

_ijms, 2020, doi:10.3390/ijms21197307_

Round 1
Reviewer 1 Report
This is a very extensive and detailed review addressing all aspects of possible immune reactions within PDAC. The review is well referenced and as such very valuable for scientists working the field. The basic mechanism of the different immune components is delineated and well explained, and their assumed role in PDAC is described. The downside of the detailed description is that it is difficult to differentiate between major mechanisms, accompanying reactions and regulatory events as well as between pro-tumorigenic activities versus anti-tumor procedures. I realize that it might be difficult to judge those with the current knowledge, however, it may help to address the following issue:
(1) Do the described immune reactions occur in all patients (simultaneously? subsequently?) or – given the heterogeneity of PDAC – are there PDAC featuring one or the other response (predominantly), let’s say ”NK-rich” versus “Th17 rich” and how is that related to the outcome.
(2) Another issue is that the Th1 versus Th2 concept as well as the N1/N2 and M1/M2 concepts are mainly studied in the mouse, whereas in humans there is a greater plasticity of the immune cells – so the differentiation is more due to a functional status as to different lineage, and is also dynamically regulated. The fact should be discussed.
(3) A further issue are the cytokines: usually they are released in a non-directional manner. Hence, of great relevance are the corresponding receptors, and in that regard it is of interest that tumor cells may acquire receptors for cytokines. Thus, expression on tumor cells of cytokine receptors might be more relevant than the mere presence of cytokines. Also of note, there not really”pro” or “anti” inflammatory cytokines, because the response vary depending on the target cells and the surrounding micro milieu.
Author Response
Reviewer 1:
Point 1: Do the described immune reactions occur in all patients (simultaneously? subsequently?) or – given the heterogeneity of PDAC – are there PDAC featuring one or the other response (predominantly), let’s say ”NK-rich” versus “Th17 rich” and how is that related to the outcome.
NK-rich: The complex interactions of NK cells with the immunosuppressive TME of PDAC requires further research and most of the available data are based on the investigation of peripheral NK cells and not on the analysis of tumor infiltrating NK cells. However, it is known that the absolute number of NK cells in the circulation correlates positively with the survival of PDAC patients, whereas data on tumor infiltrating NK cells are not available. Generally, NK cell infiltration in PDAC tissue is low, but the in vitro proliferation capacity of isolated tumor infiltrating NK cells was associated with the overall survival. To clarify this point, a sentence on this issue is included at line 238.
Th17-rich: Probably in some tumors there is an early shift towards Th17 presence, because Th17 are detectable in PanIN lesions and are thought to initiate carcinogenesis. Furthermore, oncogenic Kras in tumor cells mediates the expression of the receptor for IL-17RA, meaning that tumor cells already at the stage of Kras mutation are responsive to IL-17. IL-17A regulates stemness of tumor cells via DCLK1 and POU2F3, which in patients correlate with worse prognosis. Considering that NK presence is rather beneficial in PDAC, we speculate that there is probably beneficial NK-rich tumor environment, whereas a shift towards Th17 responses is detrimental. However, to our knowledge, correlations between Th17 and NK cells have not been done in PDAC.
Point 2: Another issue is that the Th1 versus Th2 concept as well as the N1/N2 and M1/M2 concepts are mainly studied in the mouse, whereas in humans there is a greater plasticity of the immune cells – so the differentiation is more due to a functional status as to different lineage, and is also dynamically regulated. The fact should be discussed.
Th1/Th2: In the Th1/Th2 section we described the shift towards Th2 (GATA3+ T cells) which was detected in patients with PDAC, and also in serum of patients with respectable pancreatic adenocarcinoma the level of Th2 cytokine profile was increased, arguing for the pathogenic Th2 contribution also in humans.
N1/N2: The situation with human N1/N2 phenotypes in TANs has been discussed in the manuscript (line 335) and a new reference has been added [119].
Point 3: A further issue are the cytokines: usually they are released in a non-directional manner. Hence, of great relevance are the corresponding receptors, and in that regard it is of interest that tumor cells may acquire receptors for cytokines. Thus, expression on tumor cells of cytokine receptors might be more relevant than the mere presence of cytokines. Also of note, there not really ”pro” or “anti” inflammatory cytokines, because the response vary depending on the target cells and the surrounding micro milieu.
We agree with reviewer 1 in that cytokine receptors are crucial in mediating the effects of cytokines and that the presence or absence of the receptors dictates the overall responsiveness of the system. However, we think that the readership of this article is aware of this situation and therefore specific mentioning in the text was not chosen (Here, we do not think that reviewer 1 actually refers to processes such as trogocytosis with “acquisition of receptors” as we are not aware of studies to demonstrate this process in pancreatic cancer cells).
With respect to pro/anti-inflammatory cytokines, we have tried to keep the usage of these terms to a minimum throughout the text.
Reviewer 2 Report
I read with interest the manuscript entitled: “The Immune Microenvironment in Pancreatic Cancer”. The authors provide a detailed description of PDAC TME, focusing on immune cell compartment witgh special focus on stromal cells (eg CAFs) that definitely needed a thorough discussion, due to the critical role in shaping the immunophenotype of PDAC. I found the review well-written and up-to-date with the most recently scientific advances made in the field in the last years. The authors drawn detailed picture of the biology of the TME cells.
However, I have some comments and suggestions.
- MAJOR COMMENTS
Please add a “perspective” or inferences in “Conclusions” section.
Sometimes, the article is a bit too redundant and provides some very general, basic introductions of cell biology that are not related to PDAC, and could maybe get shortened.
Also, I think that some modifications could be made to have a more-complete overview of PDAC’s TME and some articles could be addressed.
- “1. Pancreatic cancer: clinical situation”: Could you elaborate on the Moffitt classification for normal and activated-stroma subtypes? You should comment how different cell populations (and their subtypes, eg different subtypes of CAFs) are represented in the different PDAC subtypes. Moffitt RA, Marayati R, Flate EL, et al. Virtual microdissection identifies distinct tumor- and stroma-specific subtypes of pancreatic ductal adenocarcinoma. Nat Genet. 2015;47(10):1168-1178. doi:10.1038/ng.3398
Adding to the already cited articles, there was another work (September 2020, Annals of Surgery) that supports the feasibility of patient-derived organoids for pharmacotyping.
Seppälä, Toni T. MD, PhD∗,†; Zimmerman, Jacquelyn W. MD, PhD‡,¶; Sereni, Elisabetta MD∗; Plenker, Dennis PhD§; Suri, Reecha MS∗; Rozich, Noah MD∗; Blair, Alex MD∗; Thomas, Dwayne L. II BS, BA∗; Teinor, Jonathan BS∗; Javed, Ammar MD∗; Patel, Hardik M Pharm§; Cameron, John L. MD∗,¶; Burns, William R. MD∗,¶; He, Jin MD∗,¶; Tuveson, David A. MD, PhD§; Jaffee, Elizabeth M. MD‡,¶; Eshleman, James MD, PhD¶,||; Szabolcs, Annamaria PhD∗∗; Ryan, David P. MD∗∗; Ting, David T. MD∗∗; Wolfgang, Christopher L. MD, PhD∗,¶; Burkhart, Richard A. MD∗,¶ Patient-derived Organoid Pharmacotyping is a Clinically Tractable Strategy for Precision Medicine in Pancreatic Cancer, Annals of Surgery: September 2020 - Volume 272 - Issue 3 - p 427-435 doi: 10.1097/SLA.0000000000004200
- “2.1 The role of macrophages in PDAC”: Besides works by Liou et al, Nywening et al, and Griesmann et al, there is another recent paper that analyzed particular changes along progression of pancreatic cancer (using organoid models) by Filippini et al. dealing with M2-macrophages, CD8 infiltration, T cell exclusion. Filippini, D., Agosto, S.D., Delfino, P. et al. Immunoevolution of mouse pancreatic organoid isografts from preinvasive to metastatic disease. Sci Rep 9, 12286 (2019). https://doi.org/10.1038/s41598-019-48663-7
- “2.2. The role of myeloid-derived suppressor cells”: Speaking about MDSCs immunosuppressive roles, Trovato et al. found that human M-MDSCs can mediate immunosuppression and possess specific genetic and phenotypic features.Trovato R, Fiore A, Sartori S, et al Immunosuppression by monocytic myeloid-derived suppressor cells in patients with pancreatic ductal carcinoma is orchestrated by STAT3 Journal for ImmunoTherapy of Cancer 2019;7:doi: 10.1186/s40425-019-0734-6
A recent work by Choueiry et al. demonstrated that CD200 promotes immunosuppression in PDAC (studied both in murine models and humans) through regulation of MDSCs in the stroma, and its targeting could improve immunotherapeutic strategies.
Choueiry F, Torok M, Shakya R, et al. CD200 promotes immunosuppression in the pancreatic tumor microenvironment. J Immunother Cancer. 2020;8(1):e000189. doi:10.1136/jitc-2019-000189
- “3. Contribution of Natural killer (NK) cells in PDAC”: it could be worth addressing a recent preclinical work with CD40 agonist combined with IL15 by Van Audenaerde et al. The combo yieled promising results, and these seem to be related to CD8 T cells and NK cells (which are promoted by IL15, as you previously mentioned).
Van Audenaerde, J.R., Marcq, E., von Scheidt, B., Davey, A.S., Oliver, A.J., De Waele, J., Quatannens, D., Van Loenhout, J., Pauwels, P., Roeyen, G., Lardon, F., Slaney, C.Y., Peeters, M., Kershaw, M.H., Darcy, P.K. and Smits, E.L. (2020), Novel combination immunotherapy for pancreatic cancer: potent anti‐tumor effects with CD40 agonist and interleukin‐15 treatment. Clin Transl Immunol, 9: e1165. doi:10.1002/cti2.1165
- “ Contribution of Neutrophils cells in PDAC”: in addition to the already cited work, another recent paper highlighted that NETosis (induced by IL17 axis) may be responsible for resistance to immune checkpoint inhibitors.
Zhang Y, Chandra V, Riquelme Sanchez E, et al. Interleukin-17-induced neutrophil extracellular traps mediate resistance to checkpoint blockade in pancreatic cancer. J Exp Med. 2020;217(12):e20190354. doi:10.1084/jem.20190354
- “4.3. Th17 and Treg responses in PDAC”: You should add a brief sentence explaining what PanINs are. (line 459)
- “4.4. Regulatory T cells (Tregs) and PDAC”: Changes in TME are also seen in other premalignant lesions, ie IPMNs. Mainly, Tregs and CD8 are altered. IPMN microenvironment was summarised by a recent paper by Nasca et al. Nasca, V.; Chiaravalli, M.; Piro, G.; Esposito, A.; Salvatore, L.; Tortora, G.; Corbo, V.; Carbone, C. Intraductal Pancreatic Mucinous Neoplasms: A Tumor-Biology Based Approach for Risk Stratification. Int. J. Mol. Sci. 2020, 21, 6386.
Further expanding this interesting discussion of T-regs in PDAC, there was a recent work (worth addressing) by Sivakumar et al. that, through cyTOF and scRNAseq of human PDAC, shed more light on T cell landscape, with previously unreported results on T regs, senescent T cells and T cell exhaustation. Immune responses in pancreatic cancer may be restricted by prevalence of activated regulatory T-cells, dysfunctional CD8+ T-cells, and senescent T-cells. Shivan Sivakumar, Enas Abu-Shah, David J Ahern, Edward H Arbe Barnes, Nagina Mangal, Srikanth Reddy, Aniko Rendek, Alistair Easton, Elke Kurz, Michael Silva, Lara R Heij, Zahir Soonawalla, Rachael Bashford-Rogers, Mark R Middleton, Michael L Dustin bioRxiv 2020.06.20.163071; doi: https://doi.org/10.1101/2020.06.20.163071
- “5. Cancer-associated fibroblasts (CAFs) as immunological modulators in PDAC”: this is a very general - maybe a little redundant intro - yet it lacks the immunosuppressive functions exerted by CAFs In my opinion, you should add them and maybe rephrase these few lines. A Single-Cell Window into Pancreas Cancer Fibroblast Heterogeneity ad I. Belle and David G. DeNardo Cancer Discov August 1 2019 (9) (8) 1001-1002; DOI: 10.1158/2159-8290.CD-19-0576
line 672: “Interestingly, this reflects.. etc”. I’m not sure this is relevant to the discussion.
- MINOR COMMENTS
Below are some modifications for some grammar and syntax mistakes, in order to make the text more readable.
- line 26 (Abstract): “In can be hoped that” should be changed in “It is to be hoped that/Hopefully”.
- line 62 (Pancreatic cancer: clinical situation): “check point inhibitor therapy” -> “immune checkpoint inhibitors”.
- line 63: “which however, concern” -> “which, however, concern”
- line 68: “is not yet standard clinical practice” -> “has not entered routine clinical practice yet”.
- line 76: “yet” usually goes at the end of the sentence
- line 108 (Role of innate immune cells): “environmental clues such as..” -> “environmental clues, such as…”
- line 130: “orthotopic injection mouse models” -> “orthotopic mouse models”
- line 174: very minor formatting corrections for references. Also elsewhere in the paper
- line 185 (Contribution of NK cells in PDAC): “demonstrated in different experimental mouse models”. Citation needed
- line 209: “but the cytotoxicity of PDAC-associated NK cells… etc”; minor punctuation and syntactic modifications needed as the sentence seems a bit confused (and long).
-line 267: “3. Contribution of Neutrophils cells in PDAC” -> “4. Etc” (there are 2 n.3 in the list) - line 271 (Contribution of Neutrophils in PDAC): “half-live”-> “half-life”
- line 273: I’d rather write PMNs (instead of PMN)
- line 285: “inflammation driven” -> “inflammation-driven”
- line 324: add a “,” between “tumor cells” and “PDE4D”
- line 347: “NETs contribute to mts”. Reference needed
- line 486: “by both, pancreatic cancer and T cells etc” -> “, by both pancreatic cancer and T cells, etc”
- line 549: “Tumor cell intrinsic factors” -> “Tumor cell-intrinsic factors”
- line 584: “more distant” -> “more distantly”
- line 594: as a general rule, genes go in italics. Proteins and similars go in regular form. Make corrections also elsewhere
- line 621: “contribute the tumoral” -> “contribute to the tumoral”
- line 637: “impressive” -> “impressively”
- line 677: “… in several istances”. Add citation
- line 685: “a process, which” -> “a process which”
Author Response
Reviewer 2:
Point 1: Please add a “perspective” or inferences in “Conclusions” section.
As suggested, we have included a perspective paragraph into the final Conclusions section (line 825).
Point 2: Pancreatic cancer: clinical situation: Could you elaborate on the Moffitt classification for normal and activated-stroma subtypes? You should comment how different cell populations (and their subtypes, eg different subtypes of CAFs) are represented in the different PDAC subtypes.
Adding to the already cited articles, there was another work (September 2020, Annals of Surgery) that supports the feasibility of patient-derived organoids for pharmacotyping.
We have added a new paragraph on this point at line 65. The reference of Seppälä et al. was included in the manuscript.
Point 3: The role of macrophages in PDAC: Besides works by Liou et al, Nywening et al, and Griesmann et al, there is another recent paper that analyzed particular changes along progression of pancreatic cancer (using organoid models) by Filippini et al. dealing with M2-macrophages, CD8 infiltration, T cell exclusion. The reference has been included.
Point 4: The role of myeloid-derived suppressor cells: Speaking about MDSCs immunosuppressive roles, Trovato et al. found that human M-MDSCs can mediate immunosuppression and possess specific genetic and phenotypic features.
The reference has been included.
Point 5: A recent work by Choueiry et al. demonstrated that CD200 promotes immunosuppression in PDAC (studied both in murine models and humans) through regulation of MDSCs in the stroma, and its targeting could improve immunotherapeutic strategies.
The reference and a novel sentence (line 169) have been included.
Point 6: Contribution of Natural killer (NK) cells in PDAC: it could be worth addressing a recent preclinical work with CD40 agonist combined with IL15 by Van Audenaerde et al. The combo yielded promising results, and these seem to be related to CD8 T cells and NK cells (which are promoted by IL15, as you previously mentioned).
The reference and a novel sentence (line 263) have been included.
Point 7: Contribution of Neutrophils cells in PDAC: in addition to the already cited work, another recent paper (Zhang et al.) highlighted that NETosis (induced by IL17 axis) may be responsible for resistance to immune checkpoint inhibitors.
The reference and a novel sentence (line 375) have been included.
Point 8: Th17 and Treg responses in PDAC: You should add a brief sentence explaining what PanINs are.
As suggested, a short explanation on PanINs has been included (line 483).
Point 9: Regulatory T cells (Tregs) and PDAC: Changes in TME are also seen in other premalignant lesions, ie IPMNs. Mainly, Tregs and CD8 are altered. IPMN microenvironment was summarized by a recent paper by Nasca et al.
The reference and a novel sentence (line 518) have been included.
Point 10: Further expanding this interesting discussion of T-regs in PDAC, there was a recent work (worth addressing) by Sivakumar et al. that, through cyTOF and scRNAseq of human PDAC, shed more light on T cell landscape, with previously unreported results on T regs, senescent T cells and T cell exhaustation. Immune responses in pancreatic cancer may be restricted by prevalence of activated regulatory T-cells, dysfunctional CD8+ T-cells, and senescent T-cells.
While we appreciate the work of Sivakumar and colleagues (bioRxiv), we have decided NOT to include it in our manuscript as long as it has not been subject to a peer-review process.
Point 11: Cancer-associated fibroblasts (CAFs) as immunological modulators in PDAC: this is a very general - maybe a little redundant intro - yet it lacks the immunosuppressive functions exerted by CAFs In my opinion, you should add them and maybe rephrase these few lines. A Single-Cell Window into Pancreas Cancer Fibroblast Heterogeneity ad I. Belle and David G. DeNardo Cancer Discov August 1 2019 (9) (8) 1001-1002; DOI: 10.1158/2159-8290.CD-19-0576.
We believe that the background on CAFs is justified to understand the different subtypes and the subsequent description of their impact on the tumor immune microenvironment. The reference has been added.
Point 12: “Interestingly, this reflects.. etc”. I’m not sure this is relevant to the discussion.
We understand the view of reviewer 2, but still think that some side information might be interesting to certain readers. Furthermore, in the specific case, it exemplifies that the physiological function of a given cancer protein (i.e. LIF) is high jacked in oncogenesis and that oncogenic properties are often the “re-usage” of already available functions.
Point 13: List of minor comments (not all shown).
The comments on grammar and wording have been addressed.